# Molecular Pathogenesis of Colorectal Cancer: Impact of Oncogenic Targets Regulated by Tumor Suppressive *miR-139-3p*

**DOI:** 10.3390/ijms231911616

**Published:** 2022-10-01

**Authors:** Ryutaro Yasudome, Naohiko Seki, Shunichi Asai, Yusuke Goto, Yoshiaki Kita, Yuto Hozaka, Masumi Wada, Kan Tanabe, Tetsuya Idichi, Shinichiro Mori, Takao Ohtsuka

**Affiliations:** 1Department of Digestive Surgery, Breast and Thyroid Surgery, Graduate School of Medical and Dental Sciences, Kagoshima University, Kagoshima 890-8520, Japan; 2Department of Functional Genomics, Graduate School of Medicine, Chiba University, Chiba 260-8670, Japan

**Keywords:** microRNA, expression signature, *miR-139-3p*, tumor suppressor, colorectal cancer, *KRT80*, *HK2*, *AKT*

## Abstract

We recently determined the RNA sequencing-based microRNA (miRNA) expression signature of colorectal cancer (CRC). Analysis of the signature showed that the expression of both strands of pre-*miR-139* (*miR-139-5p*, the guide strand, and *miR-139-3p*, the passenger strand) was significantly reduced in CRC tissues. Transient transfection assays revealed that expression of *miR-139-3p* blocked cancer cell malignant transformation (e.g., cell proliferation, migration, and invasion). Notably, expression of *miR-139-3p* markedly blocked RAC-alpha serine/threonine-protein kinase (AKT) phosphorylation in CRC cells. A combination of in silico database and gene expression analyses of *miR-139-3p*-transfected cells revealed 29 putative targets regulated by *miR-139-3p* in CRC cells. RNA immunoprecipitation analysis using an Argonaute2 (AGO2) antibody revealed that *KRT80* was efficiently incorporated into the RNA-induced silencing complex. Aberrant expression of Keratin 80 (*KRT80*) was detected in CRC clinical specimens by immunostaining. A knockdown assay using small interfering RNA (siRNA) targeting *KRT80* showed that reducing *KRT80* expression suppressed the malignant transformation (cancer cell migration and invasion) of CRC cells. Importantly, inhibiting *KRT80* expression reduced AKT phosphorylation in CRC cells. Moreover, hexokinase-2 (HK2) expression was reduced in cells transfected with the *KRT80* siRNAs or *miR-139-3p*. The involvement of miRNA passenger strands (e.g., *miR-139-3p*) in CRC cells is a new concept in miRNA studies. Our tumor-suppressive miRNA-based approach helps elucidate the molecular pathogenesis of CRC.

## 1. Introduction

According to the World Health Organization (Globocan 2020), colorectal cancer (CRC) is the third most common cancer (over 1,800,000 cases) worldwide and the second leading cause of cancer-related deaths (over 880,000 deaths) [1]. In clinical practice, the prognosis of CRC is relatively good if diagnosed early. However, the prognosis is consistently poor in advanced cases, with a 5-year survival rate of approximately 14% (stage III or stage IV metastatic disease) [2]. At the time of the initial diagnosis, approximately 14–18% of patients with CRC have metastases, and the treatment strategies for unresectable cases are limited [3].

The oncogenesis of CRC is illustrated by a well-known multistep model of cancer cells [4,5,6]. From previous studies, mutations in various genes involved in the oncogenesis of CRC (e.g., *APC, TP53, SMAD4, KRAS*, and *PIK3CA*) and activation of cancer signaling pathways (e.g., WNT, RAS/MAPK, PI3K, TGF-β, P53, and DNA mismatch-repair) caused by these gene mutations have been reported [4]. CRC cells have highly heterogeneous properties, requiring new therapeutic parameters for CRC from genetic and genomic points of view. As a result of this molecular heterogeneity, recent genome-wide transcriptome analyses have revealed that CRC cells can be molecularly classified into four consensus molecular subtypes (CMS1 to CMS4) [7]. The future treatment strategies for CRC patients will be based on these subtypes.

As a result of the Human Genome Project, it has become clear that a vast number of functional non-coding RNA molecules (ncRNAs) are present in the human genome [8]. Current studies have shown that numerous ncRNAs play important roles in various biological activities such as the stabilization of RNA molecules and regulation of gene expression and the cell cycle [9,10]. Extensive research to date has revealed that ncRNA dysregulation is deeply involved in the initiation and development of human diseases, including cancer [11].

Among ncRNAs, microRNAs (miRNAs), consisting of only 19–22 nucleotides, have been well studied in cancer research fields. They function as fine-tuners of gene expression control in a sequence-dependent manner [12]. A single miRNA controls numerous genes, and in turn, a single gene is controlled by numerous miRNAs [13]. As a result, miRNAs and their target genes form a very complex network within cells, and it is easy to imagine that aberrant expression of miRNAs disrupts this RNA network. Many studies have shown aberrant expression of miRNAs in CRC cells, and these miRNAs act as oncogenes and/or tumor suppressors by targeting cancer-related genes in CRC cells [14,15,16].

More recently, to identify aberrantly expressed miRNAs in CRC cells, we determined the miRNA expression signature of CRC by RNA sequencing [17]. Our signature revealed that both the guide and passenger strands of 16 miRNAs (e.g., *miR-9*, *miR-28*, *miR-29c*, *miR-30a*, *miR-99a, miR-100*, *miR-125b*, *miR-129*, *miR-133a*, *miR-139*, *miR-143*, *miR-145*, *miR-218*, *miR-195*, *miR-490*, and *miR-497*) derived from pre-miRNAs were downregulated [17]. Our recent studies showed that some passenger strands of miRNAs (e.g., *miR-30a*, *miR-99a*, *miR-143*, *miR-145*, and *miR-490*) act as tumor-suppressive miRNAs in a wide range of cancers [17,18,19,20,21]. Interestingly, the genes regulated by a specific miRNA differ depending on the type of cancer.

In this study, we focused on *miR-139-3p* (the passenger strand of pre-*miR-139*) and investigated its functional significance and target oncogenes in CRC cells. Notably, ectopic expression of *miR-139-3p* markedly blocked the phosphorylation of RAC-alpha serine/threonine-protein kinase (AKT) in CRC cells. Our search strategy for miRNA targets revealed a total of 29 genes as putative candidate targets of *miR-139-3p* in CRC cells. Of these, keratin 80 (*KRT80*) was found to be a direct target of *miR-139-3p*, and its aberrant expression enhanced the malignant transformation of CRC cells. Involvement of the passenger strand of miRNA and its gene targets in CRC pathogenesis is a new concept and provides novel insights into the molecular pathogenesis of CRC.

## 2. Results

### 2.1. Expression of miR-139-5p and miR-139-3p in CRC Specimens

Recently, we determined the miRNA expression signature of CRC by miRNA sequencing using CRC clinical specimens (GEO accession number: GSE183437). Analysis of the signature showed that 84 miRNAs were upregulated, and 70 were downregulated in CRC tissues (Figure 1A). Among downregulated miRNAs in CRC tissues, we focused on *miR-139-5p* (the guide strand) and *miR-139-3p* (the passenger strand), because both strands of miRNAs derived from pre-*miR-139* were significantly downregulated in CRC tissues. Our interest is to clarify how the passenger strand of miRNA is involved in the malignant transformation of CRC cells. The mature sequences of the two microRNAs are shown in Figure 1B.

CRC tissues and noncancerous tumor-adjacent tissues (27 paired) were used to verify the expression status of *miR-139-5p*, *miR-139-3p*, and their target genes. Clinical information of these specimens is shown in Appendix A. The expression levels of *miR-139-5p* (*p* < 0.001) and *miR-139-3p* (*p* < 0.001) were significantly lower in CRC tissues than normal tissues (Figure 1C). Next, we examined the expression levels of *miR-139-5p* and *miR-139-3p* in two CRC cell lines, HCT116 and DLD-1. In these cell lines, the expression levels of *miR-139-5p* and *miR-139-3p* were lower than those in normal epithelial tissues (Figure 1C).

Furthermore, a positive correlation was detected between *miR-139-5p* and *miR-139-3p* expression levels by Spearman’s rank analysis (*r* = 0.559, *p* < 0.001; Figure 1D).

### 2.2. Ectopic Expression Assays of miR-139-5p and miR-139-3p in CRC Cell Lines

To investigate the tumor-suppressive functions of *miR-139-5p* and *miR-139-3p*, we ectopically expressed mature *miR-139-5p* and *miR-139-3p* in two CRC cell lines, HCT116 and DLD-1, and performed functional assays, e.g., cancer cell proliferation, migration, and invasion. After *miR-139-5p* transfection, cancer cell migration in both cell lines was significantly inhibited (Figure 2A–C). In contrast, the malignant phenotypes of cancer cells, e.g., proliferation, migration, and invasion, were significantly reduced by *miR-139-3p* transfection in both cell lines (Figure 2A–C). Representative images from the migration and invasion assays are shown in Appendix A.

### 2.3. Identification of Oncogenes Regulated by miR-139-3p in CRC Cells

Based on these expressions and functional analysis, aberrant expression of *miR-139-3p* and disruption of its gene regulation mechanisms were considered to be more deeply involved in the malignant pathogenesis of CRC. We focused on *miR-139-3p* (passenger strand) for further validation.

We investigated whether epigenetic modifications affect the downregulation of *miR-139-3p* in CRC cells. After treatment of Trichostatin A (TSA) in CRC cells, the expression level of *miR-139* was increased compared to TSA untreated cells (Appendix A). In addition, *miR-139-3p* expression level was elevated by 5-aza-2-deoxycytidine (5-aza-dC) treatment in CRC cells (Appendix A). These results suggest that histone deacetylation and DNA methylation are closely involved in the downregulation of *miR-139-3p* in CRC cells.

The following hypotheses regarding *miR-139-3p* target genes in CRC cells were made: the target genes of *miR-139-3p* have one or more binding site(s), are downregulated after *miR-139-3p* transfection in CRC cells, and are upregulated in CRC tissues. We combined the gene expression data from two databases (TargetScan and GEPIA2) with gene expression data from *miR-139-3p*-transfected CRC cells (GSE155659) to search for genes that meet these three criteria. A flowchart of the search strategy is shown in Figure 3. A total of 95 putative targets of *miR-139-3p* in CRC cells were identified.

We assessed the expression levels of putative *miR-139-3p* target genes in CRC clinical tissues using The Cancer Genome Atlas database via the GEPIA2 platform. A total of 29 genes were significantly upregulated in CRC clinical specimens (colon adenocarcinoma or rectal adenocarcinoma) in this database (*p* < 0.01: Table 1, Appendix A). GEPIA2 analysis revealed that the expression level of *KRT80* was fairly low in normal tissues (Appendix A). Genes expressed exclusively in cancer cells are appropriate therapeutic targets for CRC. We focused on *KRT80* in the subsequent functional analyses in CRC cells.

### 2.4. Direct Regulation of KRT80 by miR-139-3p in CRC Cells

In CRC cells transfected with *miR-139-3p*, both the mRNA and protein levels of KRT80 were significantly downregulated (Figure 4A).

Next, RNA immunoprecipitation (RIP) analysis was performed to confirm that *KRT80* mRNA was incorporated into the RNA-induced silencing complex (RISC) after *miR-139-3p* transfection. The RIP assay concept is illustrated in a schematic in Figure 4B. In samples subjected to immunoprecipitation using an Argonaute2 (AGO2) antibody, quantitative real-time reverse-transcription PCR (qRT-PCR) showed that the *KRT80* mRNA level was significantly higher than that in mock and miRNA control-transfected cells (*p* < 0.001; Figure 4B). Ago2-bound *miR-139-3p* and *KRT80* mRNA were isolated by immunoprecipitation using the AGO2 antibody, suggesting that the RISC plays a central role in miRNA biogenesis (Figure 4B).

Finally, a dual-luciferase reporter assay was performed to confirm that *miR-139-3p* binds directly to the 3′ untranslated regions (UTR) of *KRT80*. Luciferase activity was significantly reduced following co-transfection with *miR-139-3p* and a vector containing the *miR-139-3p*-binding site within the 3’-UTR of *KRT80* (Figure 4C). In contrast, co-transfection with a vector containing the *KRT80* 3’-UTR in which the *miR-139-3p*-binding site was deleted resulted in no change in luciferase activity (Figure 4C).

### 2.5. Knockdown Assays by Small Interfering RNAs (siRNAs) Targeting KRT80 in CRC Cell Lines

To assess the functional significance of *KRT80* in CRC cells, we performed knockdown assays using siRNAs corresponding to *KRT80* mRNA. First, the inhibitory effects of two different siRNAs (*siKRT80*-1 and *siKRT80*-2) targeting *KRT80* in two cell lines were examined. Both *KRT80* mRNA and protein levels were effectively suppressed after transfection of each siRNA into HCT116 and DLD-1 cells (Appendix A).

Knockdown of *KRT80* slightly inhibited cell proliferation (Figure 5A) and markedly inhibited migration and invasion in both HCT116 and DLD-1 cells (Figure 5B,C). Representative photographs from the migration and invasion assays are shown in Appendix A.

Based on the previous report that overexpression of *KRT80* induced epithelial-mesenchymal transition (EMT)-related genes and activated AKT signaling via phosphorylation of AKT (Ser 473) [22], Western blotting for phosphorylation of AKT was performed on *KRT80* and *miR-139-3p*.

Notably, transfection of the *KRT80* siRNAs suppressed the phosphorylation of AKT (Figure 5D).

In addition, expression of *miR-139-3p* markedly inhibited the phosphorylation of AKT in CRC cells, according to Western blot analysis.

### 2.6. Aberrant Expression of KRT80 Protein in CRC Clinical Specimens

Protein expression of KRT80 was assessed by immunohistochemistry in CRC clinical specimens. Overexpression of KRT80 protein was detected in cancer lesions (Figure 6).

### 2.7. KRT80-Mediated RNA Networks in CRC Cells

To explore *KRT80*-regulated RNA networks in CRC, we performed comprehensive gene expression analyses in *KRT80*-knockdown CRC cells. A total of 52 genes were identified as downregulated in both *KRT80*-knockdown CRC cell lines (log_2_ fold change < −1.0: Table 2). Our expression data were deposited in the GEO database (GEO accession number: GSE208785).

In this study, we focused on hexokinase 2 (*HK2*) because it was identified as a *miR-139-3p* target in CRC cells (Table 1). *HK2* was commonly regulated by *miR-139-3p* and *KRT80* in CRC cells (Figure 7A). Moreover, *HK2* was directly regulated by *miR-139-3p* in CRC cells, by RIP assay and dual luciferase reporter assay (Appendix A). In addition, *HK2* expression was upregulated in CRC tissues (Appendix A), and a vast number of studies showed that aberrant expression of *HK2* enhances cancer cell malignant transformation in various types of cancers. Our results showed that *HK2* expression was reduced in cells transfected with *siKRT80* (Figure 7B) or *miR-139-3p* (Figure 7C).

### 2.8. Expression of Target Genes in Clinical Specimens and Correlation

In the analysis using surgical specimens (27 paired normal and cancerous tissues), we observed marked suppression of *miR-139-3p* and marked upregulation of *KRT80* in cancer tissues (Figure 1C and Appendix A). In addition, a negative correlation was observed between the expression of *miR-139-3p* and *KRT80* in CRC specimens (Appendix A). Contrary to the TCGA data analysis, we did not find any significant upregulation of *HK2* in our cancerous samples.

## 3. Discussion

Because CRC is a heterogeneous disease, as indicated by our genome-wide transcriptome analysis, it is necessary to search for diagnostic markers and therapeutic target molecules in an individualized manner. Recently, we determined the miRNA expression signature of CRC using RNA sequencing [17]. In that study, we found that *miR-490-3p* acted as a tumor-suppressive miRNA in CRC cells, and expression of its gene targets (*IRAK1*, *FUT1,* and *GPRIN2*) was significantly predictive of 5-year overall survival in CRC patients [17]. This new miRNA expression signature of CRC will be a useful tool for elucidating the molecular pathogenesis of this disease.

Aberrant expression of miRNAs is frequently observed in several types of cancers [14,15,16]. A vast number of studies showed that epigenetic modification (histone modifications and promoter DNA methylation) is closely involved in the silencing of miRNAs expression in cancer cells [23,24,25,26]. A recent study showed that *miR-139* was epigenetically silenced by histone H3 lysine 27 trimethylation (H3K27me3) in lung cancer cells [25]. Our present data (TSA and 5-aza-dC treatment) showed that both events of histone deacetylation and DNA methylation were closely involved in the silencing of *miR-139-3p* on CRC cells. It has been shown that *miR-139-3p* silencing plays a pivotal role in human oncogenesis.

Our recent studies revealed that some passenger strands of miRNAs are closely involved in the molecular pathogenesis of a wide range of human cancers, e.g., *miR-30c-2-3p*, *miR-101-5p, miR-143-5p*, and *miR-145-3p* [19,21,27,28]. Based on our CRC signature, we focused on *miR-139-3p* (the passenger strand derived from pre-*miR-139*) in this study. We have analyzed the passenger stand *miR-139-3p* in several types of cancers and found that it acts as a tumor-suppressive miRNA in bladder cancer, renal cell carcinoma, and head and neck squamous cell carcinoma by targeting several genes closely linked to cancer pathogenesis [29,30,31]. Here, the function of *miR-139-3p* in CRC cells was clarified and found to be consistent with previous reports. As we have discussed, our in vitro assays showed that *miR-139-3p* acted as a tumor suppressive miRNA in CRC cells. However, the endogenous expression levels of passenger strands of miRNAs are little, and the full picture of the functions of passenger strands of miRNAs in vivo remains unknown. In order to investigate the in vivo functions of miRNAs, it is essential to generate and analyze cells that constitutively express miRNAs or cells in which miRNA expression is completely knocked out.

Several oncogenic signaling pathways are activated in CRC cells, of which PI3K/AKT/mTOR signaling is frequently activated [32,33,34,35]. Therefore, inhibiting activation of this signaling pathway is an attractive strategy for CRC treatment [32,36,37,38]. The AKT serine/threonine kinase is activated by phosphatidylinositol-3 kinase (PI3K) or phosphoinositide-dependent kinases via phosphorylation of Thr308 or Ser473 in AKT and activated AKT phosphorylates various downstream protein substrates (e.g., mTOR, glycogen synthase kinase 3 beta, and forkhead box protein O1) [39]. Aberrant expression and activation of AKT have been observed in many types of cancers, including CRC [40]. Notably, ectopic expression of *miR-139-3p* inhibited the phosphorylation of AKT in CRC cells in this study.

Next, we searched for target genes regulated by *miR-139-3p* in CRC cells, particularly those involved in AKT phosphorylation. A unique feature of miRNAs is that they regulate different sets of genes depending on the cancer cell type.

We identified 29 genes as tumor-suppressive targets of *miR-139-3p* in CRC cells. Of these, we focused on *KRT80* because its expression was significantly different between cancer and normal tissues. Ideally, a therapeutic target molecule for cancer is not expressed in normal cells. Expression levels of *KRT80* in normal tissues were assessed using previous large-scale transcriptional analysis data [41]. Expression of *KRT80* was detected in skin, esophagus, and salivary glands. In contrast, *KRT80* was hardly expressed in other tissues (Appendix A).

We showed that aberrant expression of *KRT80* enhanced the malignant phenotypes of cancer cells (i.e., proliferation, migration, and invasion). Interestingly, overexpression of *KRT80* induced EMT-related genes and activated the AKT signaling through phosphorylation of AKT (Ser 473) [22]. Considering our present data and previous reports, it was strongly suggested that the *miR-139-3p*/*KRT80*/p-AKT axis influences the migration and invasive abilities of CRC cells. In ovarian cancer, overexpression of KRT80 induced the expression of genes related to epithelial–mesenchymal transition and activated both MEK and ERK [42]. In gastric cancer, overexpression of the circular RNA Circ*PIP5K1A* induced expression of KRT80 and activated the PI3K/AKT pathway via *miR-671-5p* adsorption [43]. Moreover, KRT80 expression was significantly correlated with clinical parameters, such as lymph node metastasis and pathological stage, in CRC and ovarian cancer [22,42]. Together, these data suggest that *KRT80* is a potential therapeutic target for CRC.

We also investigated genes affected by *KRT80* in CRC cells. In CRC cells, the expression of several genes was suppressed after the knockdown of *KRT80* expression. Among these genes, we focused on *HK2*. The four members of the HK family (HK1-4) in mammals catalyze the conversion of glucose to glucose-6-phosphate, and they are involved in the first and rate-limiting step of glycolysis [44,45,46]. Previous studies reported that Akt and HK2 are overexpressed in cancer cells and that there is a positive correlation between activation of the PI3K/Akt/mTORC1 pathway and HK2 expression [47,48,49]. These findings indicate that simultaneous inhibition of glycolysis and the AKT/mTOR signaling pathway is effective in suppressing the growth of cancer cells [50].

## 4. Materials and Methods

### 4.1. Clinical Specimens Used to Evaluate miR-139-5p and miR-139-3p Expression

Fifty-four clinical specimens (27 CRC tissues and 27 normal colon tissues) were used to evaluate the expression status of *miR-139-5p/3p*. All specimens used in this study were obtained by surgical resection at Kagoshima University Hospital between 2014 and 2017. Normal colon tissue was collected from adjacent sites to the specimen from which each CRC tissue sample was taken. All patients provided written informed consent for the use of their specimens. This study was conducted in accordance with the guidelines of the Declaration of Helsinki and was approved by the Ethics Committee of Kagoshima University (approval number 160038 (28–65); date of 19 March 2021). The clinical information was described in our previous study [17].

### 4.2. CRC Cell Lines and Cell Culture

Two CRC cell lines, HCT116 and DLD-1, were used in this study. HCT116 cells were obtained from the RIKEN Cell Bank (Tsukuba, Ibaraki, Japan), and DLD-1 cells were obtained from the Cell Resource Center for Biomedical Research Bank (Sendai, Miyagi, Japan). HCT116 was cultured in DMEM medium supplemented with 10% concentration of fetal bovine serum (FBS), and DLD-1 was cultured in RPMI-1640 medium, also supplemented with 10% concentration of fetal bovine serum (FBS).

### 4.3. RNA Extraction and Quantitative Real-Time Reverse-Transcription PCR (qRT-PCR)

The protocols used for RNA extraction and qRT-PCR were described in our previous studies [51,52]. In brief, Total RNA was isolated from cell lines using TRIzol reagent according to the manufacturer’s protocol. RNA samples were reverse transcribed using the High-Capacity cDNA Reverse Transcription Kit (Applied Biosystems, Waltham, MA, USA). qPCR was performed using PCR Master Mix (Applied Biosystems, Waltham, MA, USA) and Fast SYBR Gren Master Mix (Applied Biosystems, Waltham, MA, USA), StepOnePlus real-time PCR system (Applied Biosystems, Waltham, MA, USA). Gene expressions were quantified relatively by the delta-delta Ct method (used *GUSB* as internal control). TaqMan assays used in this study are summarized in Appendix A. The sequences of primers for SYBR green assays are summarized in Appendix A.

### 4.4. Regulation of miR-139-3p Expression by DNA Demethylation

Cells were treated with 5-Aza-2′-deoxycytidine (5-aza-dC; Wako, Osaka, Japan) at concentrations of 0.5, 1,2,5, and 10 μmol/L for 96 h. Cells were first cultured in growth medium; after 24 h of incubation, the medium was replaced with fresh medium containing 5-aza-dC or Dimethyl sulfoxide (DMSO, negative control), and cells were incubated for another 48 h; after 48 h of treatment, the medium was again replaced with fresh medium containing 5-aza-dC or DMSO and cells were cultured for additional 48 h. After 120 h treatment, total RNA was isolated. The expression levels of *miR-139-3p* were measured by qRT-PCR.

### 4.5. Regulation of miR-139-3p Expression by Histone Deacetylation

Cells were treated with Tricostatin A (TSA; Wako, Osaka, Japan) at 0.1 or 0.5 μmol/L concentration for 24 h. Cells were first grown in growth medium. After 24 h of incubation, the medium was replaced with fresh medium containing TSA or DMSO and the cells were incubated for an additional 24 h. After 48 h treatment, total RNA was isolated. Expression levels of *miR-139-3p* were measured by qRT-PCR.

### 4.6. Transfection of miRNAs and siRNAs into CRC Cells

The protocols used for transient transfection of miRNAs and siRNAs were described in our previous studies [51,52]. The miRNA precursors and siRNAs used in this report were detailed in Appendix A. Opti-MEM (Gibco, Carlsbad, CA, USA) and Lipofectamine^TM^ RNAiMax Transfection Reagent (Invitrogen, Waltham, MA, USA) were used for miRNA and siRNA transfection of miRNAs and siRNAs into CRC cell lines. All miRNA precursors and siRNAs were transfected into the CRC cell line at 10 nM. Mock transfection consisted of cells without precursors or siRNAs. Control groups were transfected with the negative control precursor.

### 4.7. Functional Analyses (Tumor Suppression and Promotion Assays) in CRC Cell Lines

The tumor-suppressive functions of miRNAs were evaluated by transient transfection assays using mature *miR-139-5p* and *miR-139-3p*. The tumor-promoting functions of *KRT80* (loss-of-function assays) were assessed by siRNA transfection assays using siRNAs targeting *KRT80*. Functional assays (proliferation, migration, and invasion assays) were performed according to procedures of previous studies [51,52]. Briefly, for proliferation assays, HCT116 or DLD-1 cells were transferred into 96-well plates at 3.0 × 10^3^ cells/well. Cell proliferation was assessed using XTT assay kit II (Sigma-Aldrich, St. Louis, MO, USA) 72 h after the transfection procedure. For the migration and invasion assay, HCT116 and DLD-1 cells were transfected in 6-well plates at 3.0 × 10^5^ cells/well; 48 h later, transfected HCT116 and DLD-1 cells were added to each chamber at 1.0 × 10^5^ cells/well. Corning BioCoat^TM^ cell culture chambers (Corning, Corning, NY, USA) were used for the migration assay and Corning BioCoat Matrigel Invasion Chambers were used for the invasion assay. cells on the underside of the chamber membrane were stained and counted for analysis. All experiments were performed in triplicate. The details of the reagents used in these analyses are listed in Appendix A.

### 4.8. Identification of Putative Targets Regulated by miR-139-3p in CRC Cells

To identify oncogenic targets controlled by *miR-139-3p* in CRC cells, data were merged from the following sources to narrow down the targets: (1) Target Scan Human 8.0 database (http://www.targetscan.org/vert_80, accessed on 6 August 2021) [53], (2) gene expression data from *miR-139-3p* transfected CRC cells (GEO accession number, GSE155659), and (3) gene expression database from CRC clinical tissues using the GEPIA2 platform (http://gepia2.cancer-pku.cn/#index; accessed on 10 April 2022) [54].

### 4.9. RIP Assay

The assay for RIP was performed according to previous studies [55]. Briefly, CRC cells were cultured in 6-well dish at 3.0 × 10^5^/well concentration. Negative control miRNA precursors and *miR-139-3p* precursors were transfected. After 12 h, immunoprecipitation was performed using the MagCapture^TM^ microRNA Isolation Kit, Human Ago2, obtained from FUJIFILM Wako Pure Chemical Corporation (Wako, Osaka, Japan) according to the manufacturer’s protocol. Expression levels of *KRT80* and *HK2* bound to Ago2 were measured by qRT-PCR. TaqMan assays used in this study are summarized in Appendix A. The sequences of primers for SYBR green assays are summarized in Appendix A.

### 4.10. Dual-Luciferase Reporter Assay

The dual-luciferase reporter assay was performed to determine whether *miR-139-3p* binds directly to the 3′-UTR of *KRT80*. A partial wild-type sequence, including the seed sequence, of the *KRT80* 3′-UTR, was inserted into the psiCHECK-2 vector (C8021; Promega, Madison, WI, USA). Alternatively, the same *KRT80* 3′-UTR sequence but with the *miR-139-3p* binding site deleted was also inserted into the same vector to create the deletion-type construct. The design of each vector cloning sequence into wild-type and deletion-type were shown in Appendix A mRNA sequences of *KRT80* and *HK2* were cited from National Center for Biotechnology Information database [56]. The dual-luciferase reporter assay was performed according to previous studies [17,52]. The reagents used in the assay are listed in Appendix A.

### 4.11. Western Blot and Immunohistochemical Analyses

The procedures for Western blot and immunohistochemical analyses were performed according to our previous studies [51,52]. In brief, 72 h after transfection, cells were collected, and lysates were prepared. Next, 18 μg/lane of protein lysate was separated on e-PAGEL (ATTO, Tokyo, Japan), transferred to PVDF membranes, and incubated with primary antibody overnight at 4 °C and with secondary antibody for 1 h at room temperature. GAPDH was used as an internal control. The antibodies used are listed in Appendix A, and the clinical specimens evaluated by immunohistochemistry are shown in Appendix A.

### 4.12. Statistical Analyses

JMP Pro 15 (SAS Institute Inc., Cary, NC, USA) was used for the statistical analyses. Differences between two groups were assessed using Welch’s *t*-test and those among multiple groups using Dunnett’s test. Spearman’s test was used for the correlation analyses. A *p*-value less than 0.05 was considered statistically significant.

## 5. Conclusions

Based on the miRNA expression signature of CRC obtained by RNA sequencing, the expression of *miR-139-3p* (the passenger strand) was significantly reduced in CRC tissues. Functional assays revealed that expression of *miR-139-3p* attenuated cancer cell malignant phenotypes, indicating that *miR-139-3p* acts as a tumor suppressor in CRC cells. *KRT80* was identified as a direct target of *miR-139-3p*, and aberrant expression of *KRT80* was confirmed in CRC clinical specimens. Moreover, *HK2* expression was regulated by both *miR-139-3p* and *KRT80* in CRC cells. Exploration of miRNA-regulated molecular networks provides important information for identifying therapeutic targets for CRC.

## Figures and Tables

**Figure 1 ijms-23-11616-f001:**
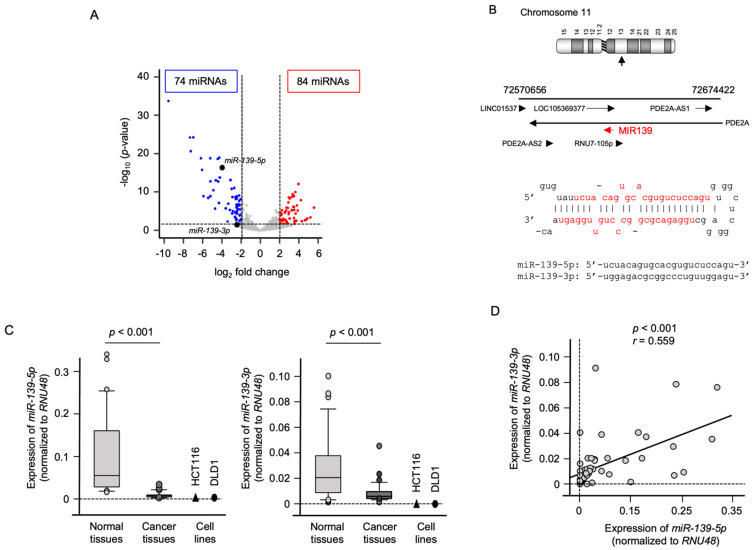
Expression of *miR-139-5p* and *miR-139-3p* in CRC clinical specimens. (**A**) Volcano plot of the miRNA expression signature based on miRNA sequencing (GEO accession number: GSE183437). The log_2_ fold-change (FC) is plotted on the *x*-axis and the log_10_ *p*-value on the *y*-axis. The blue points represent the downregulated miRNAs with log_2_ FC < −2.0 and *p* < 0.05. The red points represent the upregulated miRNAs with log_2_ FC > 2.0 and *p* < 0.05. Downregulated expressions of *miR-139-5p* and *miR-139-3p* are plotted. (**B**) Chromosomal location of *pre-miR-139* in the human genome. The mature sequences of *miR-139-5p* (the guide sequence) and *miR-139-3p* (the passenger strand) are shown. (**C**) Expression levels of *miR-139-5p* and *miR-139-3p* validated in CRC clinical specimens and CRC cell lines (HCT116 and DLD-1). The expression of both miRNAs was significantly downregulated in cancer tissues (*p* < 0.001). (**D**) Spearman’s rank test showed positive correlations between *miR-139-5p* and *miR-139-3p* expression levels in clinical specimens (*r* = 0.559, *p* < 0.001).

**Figure 2 ijms-23-11616-f002:**
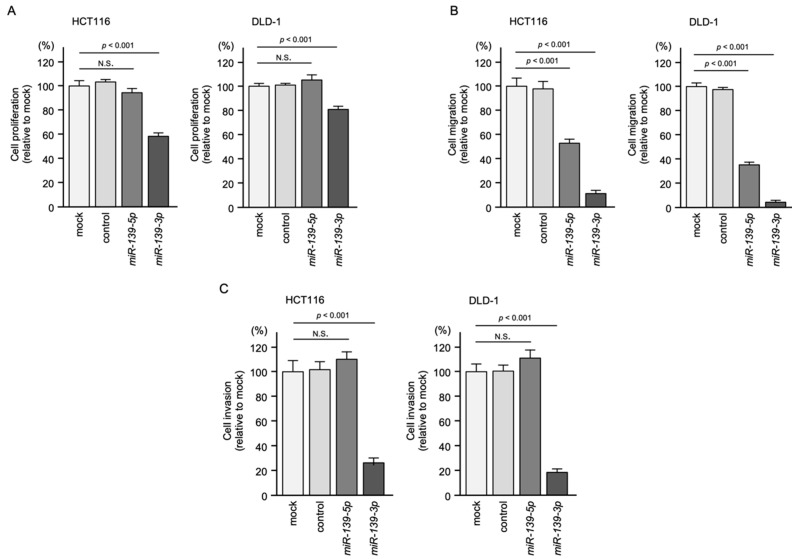
Tumor-suppressive functions of *miR-139-5p* and *miR-139-3p* in CRC cells (HCT116 and DLD-1). (**A**) Cell proliferation assessed by XTT assay. At 72 h after transient transfection of miRNAs, cancer cell viability was analyzed. (**B**) Cell migration ability assessed using a membrane culture system. At 48 h after miRNA transfection, the cells were seeded into the migration chambers. (**C**) Cell invasion ability assessed by Matrigel invasion assay. At 48 h after miRNA transfection, the cells were seeded into the invasion chambers. (N.S.: not significant compared to mock group.).

**Figure 3 ijms-23-11616-f003:**
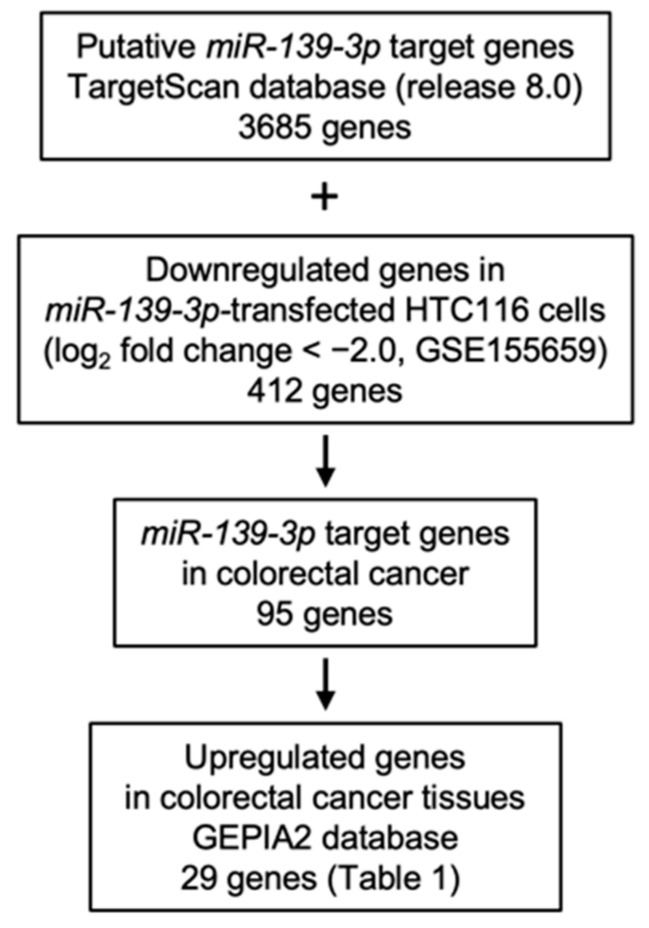
Search strategy for oncogenes regulated by *miR-139-3p* in CRC cells. To identify *miR-139-3p* targets in CRC cells, we assessed the TargetScan database and gene expression data from *miR-139-3p*-transfected HCT116 cells (GEO accession number: GSE155659). To evaluate genes upregulated in CRC clinical specimens, we used the GEPIA2 database. A total of 29 genes were identified as potential oncogenic targets regulated by *miR-139-3p* in CRC cells.

**Figure 4 ijms-23-11616-f004:**
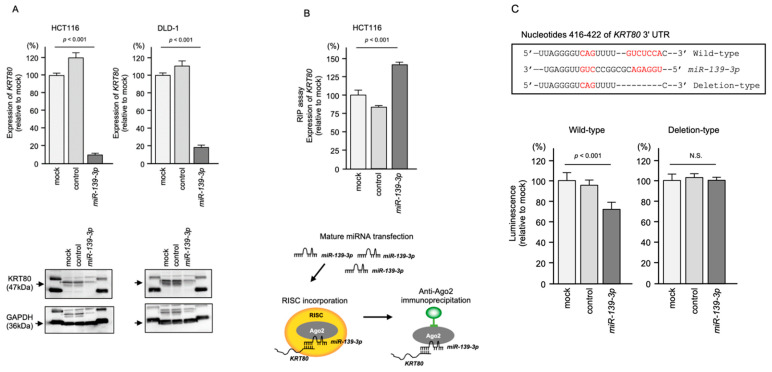
Regulation of *KRT80* expression by *miR-139-3p* in CRC cells. (**A**) Reduced mRNA and protein levels of *KRT80* in *miR-139-3p*-transfected CRC cells. At 72 h after *miR-139-3p* transfection, the cells were subjected to real-time PCR and Western blot analyses. (**B**) RNA immunoprecipitation assay of RISC-incorporated *KRT80* mRNA using an Ago2 antibody. Real-time PCR data indicated that *KRT80* mRNA was incorporated into RISC. Schematic illustration showed *miR-139-3p* and *KRT80* were incorporated into RISC. (**C**) TargetScan database analysis of the putative *miR-139-3p* binding site in the 3′UTR of *KRT80*. Dual-luciferase reporter assays showed reduced luminescence activity after co-transfection of the wild-type vector and *miR-139-3p* in HCT116 cells (left panel). Normalized data were calculated as the *Renilla/Firefly* luciferase activity ratio (N.S.: not significant compared with the mock group).

**Figure 5 ijms-23-11616-f005:**
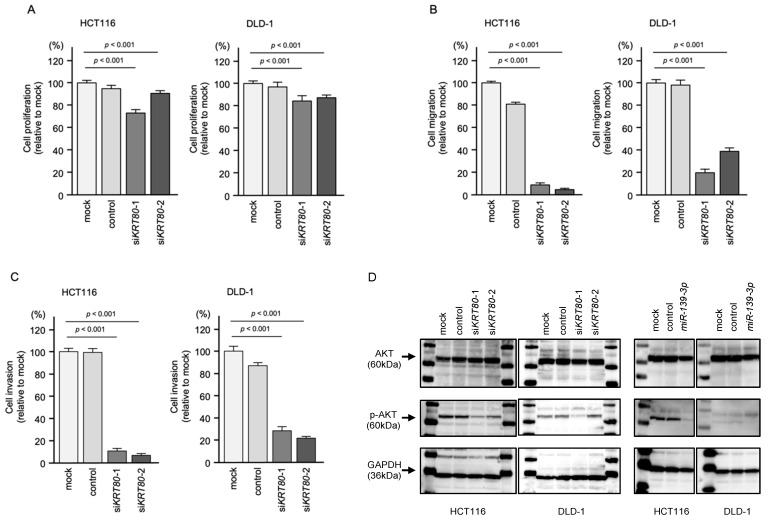
Functional assays in CRC cells after siRNA-mediated *KRT80* knockdown. (**A**) Cell proliferation assessed by XTT assay at 72 h after siRNA transfection. (**B**) Cell migration assessed using a membrane culture system at 48 h after seeding miRNA-transfected cells into the chambers. (**C**) Cell invasion assessed by Matrigel invasion assays at 48 h after seeding miRNA-transfected cells into the chambers. (**D**) Western blot analysis of AKT and phosphorylated AKT at 72 h after *siKRT80* and *miR-139-3p* transfection.

**Figure 6 ijms-23-11616-f006:**
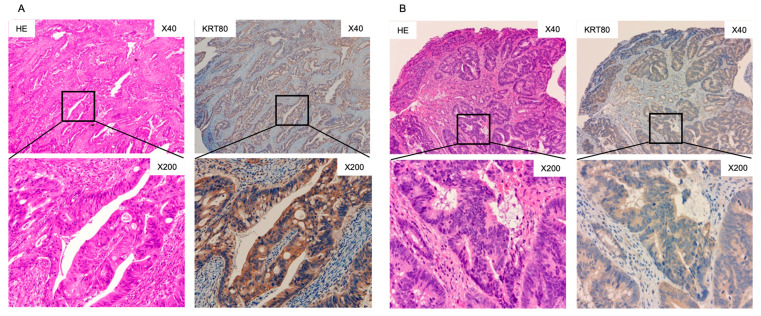
Immunohistochemical staining of KRT80 in CRC clinical specimens. Hematoxylin and eosin (HE) staining and KRT80 immunohistochemical staining in two colorectal cancer patients: (**A**) female aged 75 years with stage IV CRC, (**B**) male aged 82 years with stage IIIC CRC. The slides on the left show HE staining, and those on the right side show KRT80 immunohistochemical staining at 40× (upper) and 200× (lower) magnifications. KRT80 immunohistochemical staining showed that staining was confined to cancer tissues, with no staining in the stroma or mucus components.

**Figure 7 ijms-23-11616-f007:**
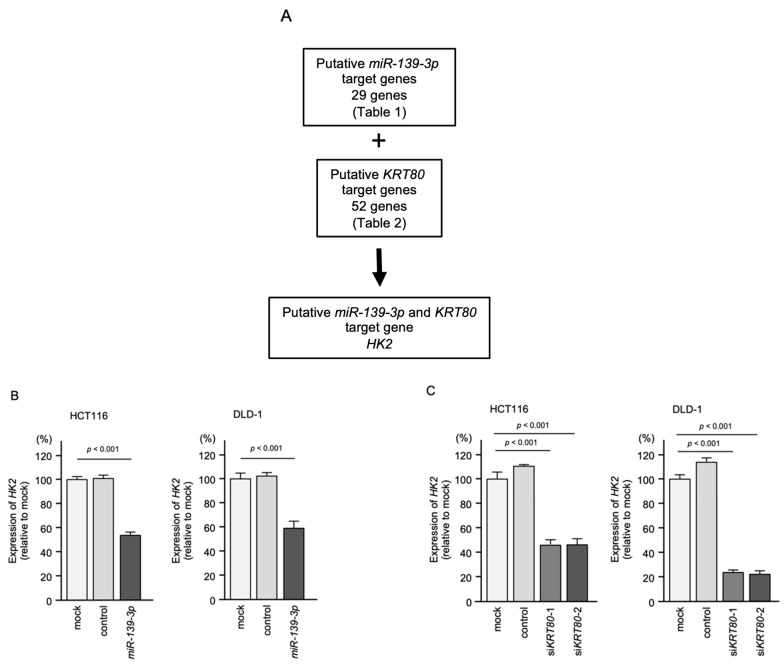
Regulation of *HK2* expression by *miR-139-3p* and *KRT80* in CRC cells. (**A**) Identification of genes commonly regulated by *miR-139-3p* and *KRT80* in CRC cells. (**B**) Reduced expression of *HK2* after *miR-139-3p* transfection in CRC cells (HCT116 and DLD-1). At 72 h after miRNA transfection, the cells were subjected to real-time PCR. (**C**) Reduced expression of *HK2* after transfection of siRNAs targeting *KRT80* in CRC cells (HCT116 and DLD-1). At 72 h after siRNA transfection, the cells were subjected to real-time PCR.

**Table 1 ijms-23-11616-t001:** Candidate gene targets of *miR-139-3p* significantly overexpressed in CRC cells.

Entrez Gene ID	Gene Symbol	Gene Name	No. of Binding Sites	*miR-139-3p*-Transfected HCT116 Cells log_2_ FC < −2
9768	*KIAA0101*	KIAA0101	2	−3.773
6541	*SLC7A1*	Solute carrier family 7 (cationic amino acid transporter, y+ system), member 1	4	−3.072
23094	*SIPA1L3*	Signal-induced proliferation-associated 1 like 3	2	−2.961
79628	*SH3TC2*	SH3 domain and tetratricopeptide repeats 2	1	−2.921
201232	*SLC16A13*	Solute carrier family 16, member 13	1	−2.905
27286	*SRPX2*	Sushi-repeat containing protein, X-linked 2	1	−2.901
118932	*ANKRD22*	Ankyrin repeat domain 22	1	−2.901
3099	*HK2*	Hexokinase 2	1	−2.729
57116	*ZNF695*	Zinc finger protein 695	2	−2.525
3352	*HTR1D*	5-hydroxytryptamine (serotonin) receptor 1D, G protein-coupled	1	−2.510
140893	*RBBP8NL*	RBBP8 N-terminal like	1	−2.509
4171	*MCM2*	Minichromosome maintenance complex component 2	1	−2.475
201266	*SLC39A11*	Solute carrier family 39, member 11	1	−2.474
90861	*HN1L*	Hematological and neurological expressed 1-like	1	−2.441
57402	*S100A14*	S100 calcium binding protein A14	1	−2.394
8884	*SLC5A6*	Solute carrier family 5 (sodium/multivitamin and iodide cotransporter), member 6	1	−2.371
55612	*FERMT1*	Fermitin family member 1	1	−2.359
9721	*GPRIN2*	G protein regulated inducer of neurite outgrowth 2	1	−2.358
54552	*GNL3L*	Guanine nucleotide binding protein-like 3 (nucleolar)-like	1	−2.252
5653	*KLK6*	Kallikrein-related peptidase 6	1	−2.251
157285	*SGK223*	Tyrosine-protein kinase sgk223	1	−2.224
144501	*KRT80*	Keratin 80	1	−2.193
154796	*AMOT*	Angiomotin	1	−2.126
9052	*GPRC5A*	G protein-coupled receptor, family C, group 5, member A	1	−2.114
54815	*GATAD2A*	GATA zinc finger domain containing 2A	1	−2.113
3898	*LAD1*	Ladinin 1	1	−2.046
414	*ARSD*	Arylsulfatase D	1	−2.023
90381	*TICRR*	TOPBP1-interacting checkpoint and replication regulator	1	−2.022
10189	*ALYREF*	Aly/REF export factor	1	−2.006

FC: fold change.

**Table 2 ijms-23-11616-t002:** Genes downregulated by *siKRT80* in CRC cells.

Entrez Gene ID	Gene Symbol	Gene Name	*siKRT80*-1-Transfected HCT116 Cells log_2_ FC < −1	*siKRT80*-2-Transfected HCT116 Cells log_2_ FC < −1
4155	*MBP*	Myelin basic protein	−4.259	−3.823
5027	*P2RX7*	Purinergic receptor P2X 7	−4.078	−1.151
7274	*TTPA*	Alpha tocopherol transfer protein	−2.983	−1.257
51339	*DACT1*	Disheveled binding antagonist of beta catenin 1	−2.745	−1.248
5163	*PDK1*	Pyruvate dehydrogenase kinase 1	−2.686	−2.183
114088	*TRIM9*	Tripartite motif containing 9	−2.641	−1.383
54434	*SSH1*	Slingshot protein phosphatase 1	−2.465	−1.668
148418	*SAMD13*	Sterile alpha motif domain containing 13	−2.444	−1.292
284716	*RIMKLA*	Ribosomal modification protein rimk like family member A	−2.323	−1.662
144501	*KRT80*	Keratin 80	−2.321	−1.878
285735	*LINC00326*	Long intergenic non-protein coding RNA 326	−2.257	−1.868
9194	*SLC16A7*	Solute carrier family 16 member 7	−2.221	−1.022
112399	*EGLN3*	egl-9 family hypoxia inducible factor 3	−2.209	−1.350
256435	*ST6GALNAC3*	ST6 N-acetylgalactosaminide alpha-2,6-sialyltransferase 3	−2.185	−1.875
4907	*NT5E*	5′-nucleotidase ecto	−2.087	−1.564
100287314	*LINC00941*	Long intergenic non-protein coding RNA 941	−2.077	−1.532
254128	*NIFK-AS1*	NIFK antisense RNA 1	−2.049	−1.204
3099	*HK2*	Hexokinase 2	−1.916	−3.518
1956	*EGFR*	Epidermal growth factor receptor	−1.872	−1.096
7378	*UPP1*	Uridine phosphorylase 1	−1.854	−1.023
51384	*WNT16*	Wnt family member 16	−1.835	−1.759
115330	*GPR146*	G protein-coupled receptor 146	−1.827	−1.710
170384	*FUT11*	fucosyltransferase 11	−1.818	−1.129
2113	*ETS1*	ETS proto-oncogene 1, transcription factor	−1.754	−1.284
861	*RUNX1*	RUNX family transcription factor 1	−1.720	−1.209
10397	*NDRG1*	N-myc downstream regulated 1	−1.666	−1.856
22989	*MYH15*	Myosin heavy chain 15	−1.634	−1.744
N.A.	*lnc-OR10H1-1*	lnc-OR10H1-1:1	−1.576	−1.277
644316	*FLJ43315*	asparagine synthetase pseudogene	−1.568	−1.405
4781	*NFIB*	nuclear factor i b	−1.565	−1.333
25886	*POC1A*	POC1 centriolar protein A	−1.549	−1.338
N.A.	*lnc-CLEC2D-7*	lnc-CLEC2D-7:1	−1.447	−1.571
51175	*TUBE1*	Tubulin epsilon 1	−1.383	−1.488
10549	*PRDX4*	Peroxiredoxin 4	−1.373	−1.158
843	*CASP10*	Caspase 10	−1.347	−1.054
3613	*IMPA2*	Inositol monophosphatase 2	−1.320	−1.712
100505933	*ADD3-AS1*	ADD3 antisense RNA 1	−1.296	−1.095
4783	*NFIL3*	Nuclear factor, interleukin 3 regulated	−1.267	−1.184
57834	*CYP4F11*	Cytochrome P450 family 4 subfamily F member 11	−1.265	−1.522
11199	*ANXA10*	Annexin A10	−1.262	−2.649
64946	*CENPH*	Centromere protein H	−1.251	−1.236
8614	*STC2*	Stanniocalcin 2	−1.243	−1.373
286144	*TRIQK*	Triple qxxk/R motif containing	−1.194	−1.351
2035	*EPB41*	Erythrocyte membrane protein band 4.1	−1.167	−1.156
28996	*HIPK2*	Homeodomain interacting protein kinase 2	−1.138	−1.166
4233	*MET*	MET proto-oncogene, receptor tyrosine kinase	−1.130	−1.247
100506211	*MIR210HG*	MIR210 host gene	−1.109	−1.795
23015	*GOLGA8A*	Golgin A8 family member A	−1.068	−1.087
23516	*SLC39A14*	Solute carrier family 39 member 14	−1.063	−1.441
84986	*ARHGAP19*	Rho gtpase activating protein 19	−1.062	−1.035
255082	*CASC2*	Cancer susceptibility 2	−1.045	−1.546
100507065	*LOC100507065*	Uncharacterized LOC100507065	−1.032	−1.425

FC: fold change, N.A.: not available.

## Data Availability

The data presented in this study are available on request from the corresponding author.

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
