# Peer review of "Molecular Pathogenesis of Colorectal Cancer: Impact of Oncogenic Targets Regulated by Tumor Suppressive miR-139-3p"

_ijms, 2022, doi:10.3390/ijms231911616_

Round 1

Reviewer 1 Report

This interesting and well-organized study explores the involvement of miR-139-5p and miR-139-3p in colorectal cancer (CRC), by combining experiments of molecular and cell biology, proteomics, immune-histochemistry and analysis of next-generation sequencing (NGS) data from public datasets. Notably, by investigating the expression of miR-139 in CRC specimens and normal controls, the Authors were able to “reconstruct” the pathway of this microRNA starting from its biogenesis, following its incorporation into the AGO2-RNA-induced silencing complex (AGO2-RISC) and identifying its molecular targets, some of them implicated in signaling pathways of crucial interest for CRC onset and progression, among whom the Keratin 80 gene (KRT80), that promotes CRC migration and invasion by interacting with PRKDC via activation of the AKT pathway. I personally appreciate the logic by which the experiments have been organized and the conclusions deduced by the data. Regarding the content of the manuscript, it is very well written, and in good English. The Introduction is clear and concise, Results are well reported and accompanied by clear and self-explanatory figures, and Discussion is well contextualizing the Author’s results with the current knowledges in the field. I think the topic itself is interesting and has the potential to attract a wide audience, generating interest in the scientific community.

With the aim of improving the quality of this work, I have some suggestion:

Major

1.      The Materials and Methods section still needs a little more work; the Authors often mention their previous studies without describing the experiments, but, in my opinion, it would be better also to include more details on the experimental procedures conducted, as this will lead the reader to better understand and evaluate the results obtained.

2.      Regarding the expression of KRT80, that the Authors indicate as a putative molecular target candidate for CRCs because of its expression that is very low in normal colorectal tissues, the expression level of this molecule in other non tumoral tissues should be reported (for example by exploring TCGA data), to determine the possible usefulness of this gene as a target and the possible tissues interested by side effects of a targeted therapy.

3.      Regarding CRC specimens and normal samples used for this study, is would be noteworthy to clarify if these normal specimens are taken from the same individual or not.  

4.      The RNA-Immunoprecipitation (RIP) experiment is missing some controls; in particular, a western blotting image showing the specific enrichment of AGO2 in the IP should be included, to show that the protein has been efficiently immunopurified and its fraction is enriched if compared to the input lysate; in analogy with this, a negative control is needed, by using a no-antibody sample or (better) a non-relevant IgG immunopurified sample, in order to substract the background and verify that enrichment observed in the expression of the microRNA of interest is real and not the result of a contamination lead by the cellular background. I suggest the Authors of repeating the experiments by including also this controls and the western blotting verification of AGO2 enrichment, as this will greatly improve the overall quality of the work.

Minor

1.      In some figures, for example as in Fig. 2D, it is not specified the cell line name.

2.      Figure could be enlarged just a little;

A list of abbreviation is missing. 

Reviewer 2 Report

In this manuscript, the authors discovered that miR-139-3p plays the suppressive role in CRC. Overexpressed miR-139-3p in colorectal cancer cell line inhibits multiple cellular functions. Besides, overexpression of miR-139-3p reduces the phosphorylation of AKT and identifies 29 putative miR-139-3p target genes. Among these putative target genes, miR-139-3p binds to the 3’-UTR of KRT80 and alters the KRT80 expression through the RISC-mediated gene regulation. Furthermore, overexpressed miR-139-3p and siKRT80 decreases the transcription of HK2. Taken together, the expression of miR-139-3p regulates the molecular pathogenesis of CRC. In general, this is a comprehensive study and the conclusion would provide for further clinical application. I am delight to read this manuscript in IJMS. However, some issues need to clarify before publish:

1.      The linkage between miR-139-ep and p-AKT is too weak. It will be more attractive if the authors can provide solid evidence or hypothesis.

2.      In clinical samples (27 paired), what is the relationship between miR-139-3p and KRT80/p-AKT/HK2 independently? Is positively correlation? It will provide the stronger clinical relevance.

3.      In detail molecular mechanism, dose miR-139-3p regulate the CRC cellular function via KRT80-HK2 axis? Or regulate the HK2 directly?

4.      In Fig.1C, the comparison of miR-139 expression of cell line and tissue in the same figure is not appropriated.

5.      If could, describe how and what mechanism to alter the expression of miR-139 in CRC.

Reviewer 3 Report

The manuscript details the characterisation of miR-139 as a tumour suppressor in the context of colorectal cancer. Results convincingly indicate miR-139 (especially miR-139-3p) can act as a general tumour suppressor when transiently overexpressed. The manuscript suggests likely direct target genes and indicates that one of these, KRT80, shows similar tumour suppressor properties when inhibited by siRNA. Whilst these basic conclusions are solid, I have severe reservations about the impact of such a study which is heavily reliant upon miRNA over-expression and ascribing a particular target gene to at least some of the properties of the miRNA whereas in reality, there are likely to be dozens if not hundreds of direct targets that contribute, including many likely to be more strongly repressed by miR-139 itself. The materials and methods are also not sufficiently detailed. In my opinion, significant work would be required before publication in this journal. Specific comments as follow:

1) Perhaps the greatest issue has been glossed over in the manuscript regarding 5p vs 3p. miR-139 is a 5p-dominant miRNA (see miRBase) however almost all of the effect concerns the 3p arm. This presumably is little expressed in many contexts. Thus, what is the relevance of this story highly expressing a version of a miRNA that is likely little expressed endogenously?

2) All miRNA experiments are over-expression. It is relatively easy to force an effect from miRNA over-expression (which is often hundreds of times more expressed than it would be at endogenous levels). Figure 4a might suggest this given that the target is dramatically downregulated by miR-139, yet the binding site itself (Fig 4c) appears modest (a 7-mer interaction site with minimal supplementary base pairing).

3) The study would also benefit significantly from replication of some key findings with miRNA inhibition (in a context in which miR-139 is endogenous expressed).

4) The materials and methods are insufficiently detailed to allow replication. How much miRNA/siRNA for example is transfected? Also, what proportion of the KRT80 3'UTR is cloned in the luciferase reporter (Fig 4c) ("a partial wildtype sequence was inserted")? 

5) The authors must be extremely clear discussing what they do and don't claim. Is the suggestion that KRT80 is the mechanism through which miR-139 exerts its tumour suppressor properties? If so, some degree of rescue experiments are required. If not, what claim is of significance? That miR-139 when expressed is a tumour suppressor (which is reported)?

6) Figure labelling: In 5D, presumably HCT116 is on the left and DLD-1 is on the right? Is it of significance to the study that only one of these cell lines displays basal AKT activity? In Fig 1B, I'm not sure exactly what has been sequenced and more detail is required to explain.

Round 2

Reviewer 2 Report

 I am delight to read this manuscript in IJMS.